# Structural basis of CXC chemokine receptor 1 ligand binding and activation

Naito Ishimoto [1,4], Jae-Hyun Park [1,4], Kouki Kawakami [2], Michiko Tajiri [3], Kenji Mizutani[1], Satoko Akashi[3], Jeremy R. H. Tame [1], Asuka Inoue [2] & Sam-Yong Park [1] ✉

Neutrophil granulocytes play key roles in innate immunity and shaping adaptive immune responses. They are attracted by chemokines to sites of infection and tissue damage, where they kill and phagocytose bacteria. The chemokine CXCL8 (also known as interleukin-8, abbreviated IL-8) and its G-protein-coupled receptors CXCR1 and CXCR2 are crucial elements in this process, and also the development of many cancers. These GPCRs have therefore been the target of many drug development campaigns and structural studies. Here, we solve the structure of CXCR1 complexed with CXCL8 and cognate G-proteins using cryo-EM, showing the detailed interactions between the receptor, the chemokine and Gαi protein. Unlike the closely related CXCR2, CXCR1 strongly prefers to bind CXCL8 in its monomeric form. The model shows that steric clashes would form between dimeric CXCL8 and extracellular loop 2 (ECL2) of CXCR1. Consistently, transplanting ECL2 of CXCR2 onto CXCR1 abolishes the selectivity for the monomeric chemokine. Our model and functional analysis of various CXCR1 mutants will assist efforts in structure-based drug design targeting specific CXC chemokine receptor subtypes.

Chemokines are small proteins, 8–12 kDa in size, secreted by cells at sites of injury or infection to attract and activate leukocytes that carry receptors to these molecules[1]. More than 50 chemokines and their 23 receptors in humans are crucial to a very wide variety of physiological and disease processes, including development, immune function, inflammation, and cancer. Many chemokines interact with more than one receptor and vice versa, making a complex signaling network[2]. Chemokines share a common fold and at least four conserved cysteine residues, but are grouped into four subfamilies (CXC, CC, C and CX₃C) based on the sequence motifs carrying signature cysteine residues in the N termini. CXC chemokines can be further subclassified as ELR⁺ or ELR⁻ by the presence or absence of the tripeptide motif glutamic acid-leucine-arginine (ELR) N-terminal to the first cysteine. ELR⁺ CXC chemokines target neutrophils and are also known as neutrophil

activating chemokines (NACs)[3]. Chemokine receptors are class A family GPCRs (G protein-coupled receptors) that recognise chemokines from different subfamilies. CXC chemokine receptors 1 and 2 (CXCR1 and CXCR2) were first described in 1991[4,5], and since then numerous studies have been conducted to investigate their therapeutic potential, for example in breast cancer[6]. CXCR1/2 are expressed by various immune and non-immune cells including neutrophils, macrophages, and endothelial cells. Both receptors bind the pro-inflammatory CXCL8, although CXCR2 interacts with a wider range of agonists.

CXCL8 is a member of the ELR⁺ CXC subfamily of chemokines, which dimerise under physiological conditions. It was one of the first chemokines to be discovered[7,8]. Formerly named neutrophil-activating factor (NAF)[9], it was found to trigger a rapid and transient rise in the

[1]Drug Design Laboratory, Graduate School of Medical Life Science, Yokohama City University, Tsurumi, Yokohama 230-0045, Japan. [2]Graduate School of Pharmaceutical Sciences, Tohoku University, Sendai 980-8578, Japan. [3]Structural Epigenetics Laboratory, Graduate School of Medical Life Science, Yokohama City University, Tsurumi, Yokohama 230-0045, Japan. [4]These authors contributed equally: Naito Ishimoto, Jae-Hyun Park. ✉e-mail: park@yokohama-cu.ac.jp

cytosolic level of free calcium in neutrophils (but not monocytes, lymphocytes, or platelets) through a GTP-binding protein[10]. Evolutionary studies suggest CXCL8 is ancient, although functional diversification has led to it playing different roles in different species[11]. With a mature form only 72 amino acids in length and readily expressible in bacteria, it has been the subject of many structural studies by NMR and crystallography. The monomer folds into a three-stranded β-sheet with an α-helix at the C-terminus[12,13]. The N-terminus is disordered, and a region of coil from residues 10–22, called the N-loop, is found just upstream of the first β-strand. CXCL8 dimerises with $K_d$ around 1–20 μM[14], the dimeric form being held together by edge-to-edge contacts between β-strands to form a single 6-stranded sheet; the main tertiary structural difference from the monomeric form is a shift of the N-loop[15]. The single-site mutation R26C produces non-dissociating CXCL8 dimers held together by a disulphide bond between the two copies of residue 26, with a structure essentially identical to the native dimer[16]. CXCL8 incapable of dimerisation has been produced by various mutations, most simply by removing the last seven residues of the protein to disrupt the dimer interface[17]. Mutants unable to switch between oligomeric states are referred to as "trapped dimer" and "trapped monomer". Previous work has revealed that monomeric CXCL8 interacts more strongly than the dimer with CXCR1 N-terminal peptide, and proposed that the dimerisation regulates binding to CXCR1 by reducing conformational flexibility of the chemokine[18]. Trapped CXCL8 monomer has been reported to bind CXCR1 ~70-fold and CXCR2 ~17-fold more tightly than a trapped dimer[16]. Nasser and colleagues found weaker selectivity, with the CXCL8 monomer binding CXCR1 approximately 6 times more tightly than the CXCL8 dimer, while CXCR2 showed a 4-fold preference for the CXCL8 monomer[19]. CXCL8 monomer was also found to elicit stronger responses, whether measured by phosphatidylinositol (PI) hydrolysis, secretion of β-hexosaminidase or chemotaxis[19]. Monomers and dimers of CXCL8 induced similarly rapid internalisation and extent of phosphorylation of CXCR2, but the monomer proved more effective with CXCR1[19].

The structure of unliganded CXCR1 in phospholipid bilayers was solved by NMR in 2012[20]. In 2015, the first crystal structure of a chemokine receptor complexed with a chemokine was solved, of human CXCR4 bound to a viral chemokine antagonist, vMIP-II[21]. A number of X-ray crystallographic structures of chemokine receptors have since been described in ref. 22–24. Structural studies of the interactions between chemokines and their receptors have shown the receptor N terminus binds to the chemokine core at an interface called chemokine recognition site 1 (CRS1), while the chemokine N terminus sits within a pocket of the receptor TM helical domain (chemokine recognition site 2, or CRS2). It has been shown by different groups that the receptor N-terminus plays an important role in chemokine specificity[25,26]. Both monomeric and dimeric forms of CXCL8 bind to CXCR2, and the structures have been solved of both complexes, with G proteins bound[27]. In the same paper, Liu and colleagues presented the crystal structure of CXCR2 bound to a small molecule allosteric antagonist, and with no ligand. Differences in the behaviour of CXCR1 and CXCR2 remain to be explained however, and we have therefore used cryo-EM to solve the structure of CXCR1 bound to CXCL8 and cognate G-proteins in order to clarify the functional contrast between these two closely related receptors.

## Results

### Monomeric and dimeric forms of CXCL8

Various forms of CXCL8 were purified for structural and biochemical studies with CXCR1/2, and ESI-MS was conducted to determine the proportions of CXCL8 monomer and dimer. In the mass spectrum of the wild-type protein under non-denaturing conditions at a concentration of 2–7 μM, dimer peaks were clearly seen, together with less intense monomer peaks (Fig. 1a). The observed masses for the dimer and monomer were 16763.13 ± 0.87 Da and 8381.46 ± 0.25 Da, respectively, which are in good agreement with the theoretical values of 16763.48 and 8381.74 Da, given two intra-chain disulphide bonds in the monomer. From the intensity of the monomer and dimer ions observed in the spectrum, the relative ratios were calculated to be monomer:dimer = 18:82. In the case of trapped dimer (CXCL8$^{R26C}$) under non-denaturing conditions, 7+, 8+, and 9+ ions of the dimer were observed, but no monomer ions (Fig. 1c). The observed molecular mass was 16655.21 ± 1.65 Da, which is consistent with the theoretical value of 16655.38 Da, assuming two intra-chain disulphide bonds. Acid-denaturation of the trapped dimer (CXCL8$^{R26C}$) did not produce monomer peaks in the ESI-mass spectrum, indicating that the dimer is held together by disulphide bonds (Fig. 1d). 5+ and 6+ ions of the trapped monomer (CXCL8$^{1-65}$) were observed both for the native and acid-denatured samples (Fig. 1e, f). The observed molecular mass was 7582.44 ± 0.98 Da under denaturing conditions, consistent with the theoretical value of 7582.854 Da, with two intra-chain disulphide bonds in each monomer. The data match earlier results showing that wild-type CXCL8 forms an equilibrium between monomeric and the dimeric forms, while the trapped dimer (CXCL8$^{R26C}$) is exclusively dimeric, and trapped monomer (CXCL8$^{1-65}$) is exclusively monomeric, in solution.

### Structures of CXCR1

CXCR1 was purified as a complex with the wild-type CXCL8$^{1-72}$ and Gαi1 protein, together with Gβ1 and Gγ2, stabilised by a single chain variable antibody fragment scFv16 (Supplementary Fig. 1). Single particle cryo-EM data collection yielded 120,631 particle classifications of 2,530,947 particles. Refinement of the model led to a map with a final nominal resolution of 3.4 Å (Supplementary Fig. 2). All component protein subunits of the complex are clearly visible in the final map, but none of the lipid molecules associated with the receptor. Some surface regions of the CXCL8 are poorly defined, but key parts of the CXCL8 including its N- and C-termini are seen clearly enough to build side-chains with confidence (Supplementary Fig. 3). Although the resolution of the final map is only 3.4 Å, many features of the model can be interpreted reliably, and are consistent with functional assays of mutants described below. The activities of wild-type and the modified CXCL8 proteins were confirmed using a NanoBiT-based G-protein dissociation assay, which showed similar results to previously reported studies on the response of CXCR1 to CXCL8 (Fig. 2b)[18]. CXCR1 shows 84% sequence identity with CXCR2 across ordered parts of the proteins, allowing ready structural comparison between the two (Supplementary Fig. 4). The overall features of the complex are in accordance with other class A GPCRs, with the C-terminal helix of Gαi inserted into the receptor transmembrane core that is opened by movement of TM6. Comparing our model with agonist-free, deactivated CXCR2 (PDB 6LFL)[27], deviations of up to about 4 Å are found, as expected, around the N-terminal (extracellular) end of TM5, and the N-terminal (cytoplasmic) end of TM6 (Supplementary Fig. 5a). The former location is found close to the agonist binding site, and the second location regulates binding of the G protein. Simple least-squared overlay of our CXCR1 model with the cryo-EM structure of CXCR2 bound to monomeric CXCL8 and Gi heterotrimer (PDB 6LFO)[27] shows an RMSD of only 0.909 Å between 271 Cα atoms of each GPCR, and no significant deviations of the helices (Supplementary Fig. 5b). Instead, larger differences are seen around ECL2 between TM4 and TM5, close to the bound CXCL8 (Fig. 2c, Supplementary Fig. 5b). Both receptors form two disulphide bonds on their external surfaces. One is formed by a cysteine just downstream of the N-terminal tail, linking it to the N-terminal end of TM7 (C30$^{N-term}$/C277$^{7.25}$ in CXCR1). The second bond (C110$^{3.25}$/C187$^{ECL2}$) holds the N-terminal end of TM3 to ECL2 (Fig. 2d). The N-terminal tail of CXCR1, like that of CXCR2, forms an extended conformation and becomes ordered through interactions with the

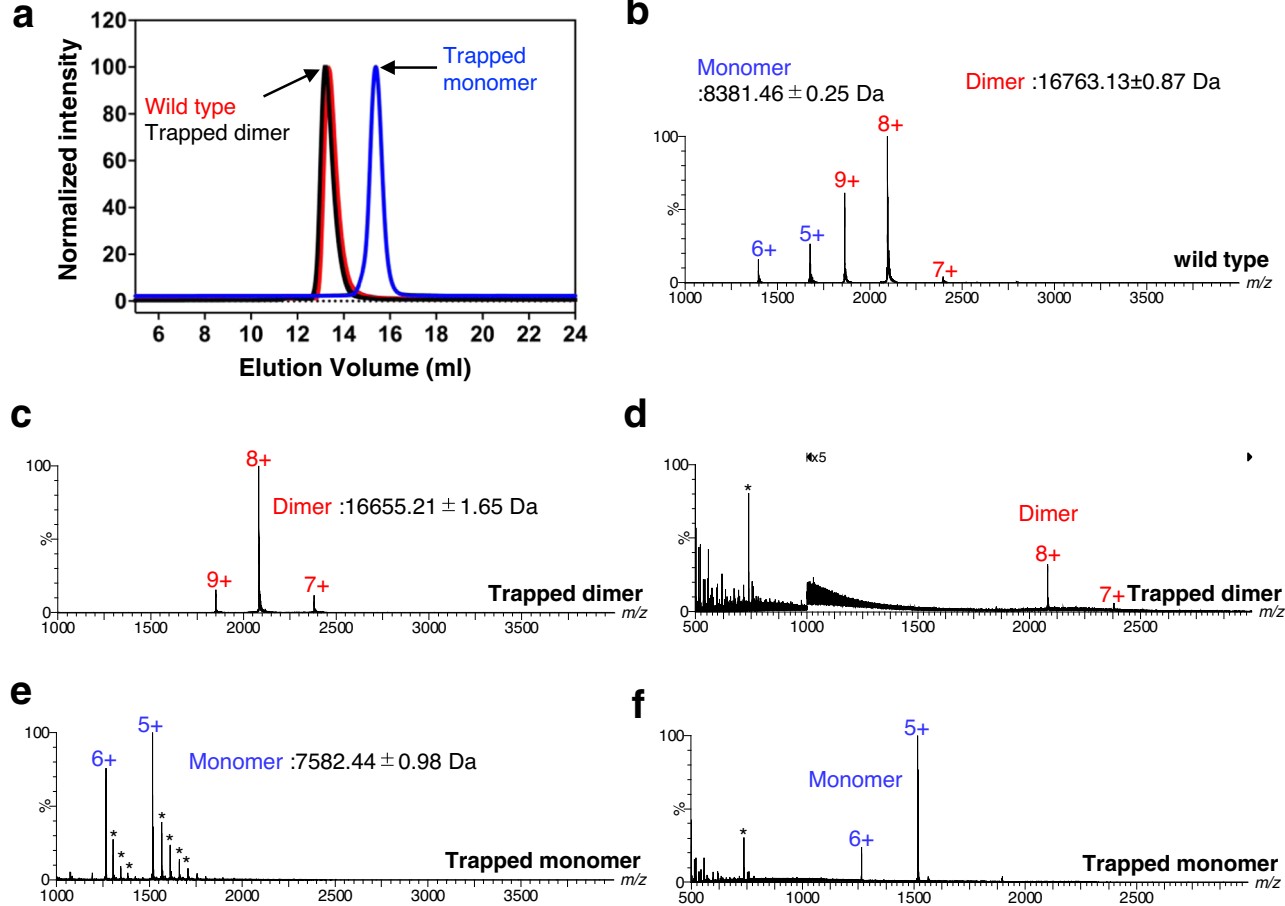

**Fig. 1 | Size exclusion chromatography and ESI mass spectrometry analysis for CXCL8. a** Size-exclusion chromatography profiles of three types of CXCL8. Curves for CXCL8 proteins are colored red (wild type), black (trapped dimer), and blue (trapped monomer), respectively. **b** Mass spectra under non-denaturing conditions for wild type CXCL8[1–72]. **c, d** Mass spectra for trapped dimer (CXCL8[R26C]) under (**c**) non-denaturing and (**d**) acid-denaturing conditions. **e, f** Mass spectra for trapped monomer (CXCL8[1–65]) under (**e**) non-denaturing and (**f**) acid-denaturing conditions. Peaks with asterisks correspond to HEPES (MW 238.3) adducts. Peaks corresponding to monomers and dimers of CXCL8 are colored blue and red, respectively.

chemokine (described below). A single copy of CXCL8 is found associated with CXCR1, with no indication of the presence of any dimer, even though the dimer interface of CXCL8 is clearly exposed to solvent (Fig. 2a, e).

**Interactions and selectivity for CXCL8**

The N-terminal region of CXCR1 fits loosely into a groove in the CXCL8 monomer to create the interaction surface CRS1 (Fig. 3a). This binding site involves both polar and apolar interactions, such as D26[N-term] making a salt bridge with K11[CXCL8], and Y27[N-term] packing against L49[CXCL8] (Fig. 3b). Rabbit CXCL8 binds the CXCR1 200-fold less potently than the human counterpart, due to two mutations in the N-terminal region, Y13[CXCL8] and K15[CXCL8] being replaced by histidine and threonine respectively in the rabbit protein[28]. The tyrosine residue is the more important of the two. In the model of CXCR1:CXCL8, Y13[CXCL8] lies between K15[CXCL8] and Y27[N-term], but K15[CXCL8] makes only van der Waals contacts with P21[N-term] and P22[N-term]. In CXCR2, Y27[N-term] is replaced by an alanine (A36[N-term]), but this is partially compensated by the introduction of L34[N-term] in CXCR2, which has no counterpart side-chain in CXCR1 (Supplementary Figs. 4, 6). Differences at CRS1 reflect the very different amino sequences of the N-terminal regions of the two receptors, which do not overlay closely. F32[N-term] of CXCR2 makes contact with several CXCL8 residues, but has no counterpart in CXCR1. Likewise, D26[N-term] of CXCR1 has no counterpart in CXCR2. Y178[ECL2] of CXCR1 contacts P32[CXCL8], but is notably replaced by valine in CXCR2, which is a much smaller side-chain (Supplementary Fig. 6).

At CRS2, the CXCL8 N terminus interacts with the surface pocket formed between the TM helices of CXCR1 (Fig. 3a). S1[CXCL8] sits between T34[N-term] and S184[ECL2] of ECL2. K3[CXCL8] is the chemokine residue that reaches most deeply into this pocket, where it lies close to D288[7.36] and E291[7.39] of TM7, and W95[2.60] of TM2 (Fig. 3c). Residues of the ELR motif also form close contacts with CXCR1, in particular E4[CXCL8] sitting near R203[5.39], R269[6.62] and R6[CXCL8], which also approaches D265[6.58] (Fig. 3c). Unlike CXCR2, our model of CXCR1 suggests a direct interaction between E4[CXCL8] and R6[CXCL8], which also brings R6[N-term] close to D265[6.58] (Fig. 3c). To confirm the importance of these interactions, we conducted the NanoBiT-Gi-dissociation assay with CXCR1 mutants, and found that the Y27[N-term]A, Y188[45.51]A, T195[5.31]A, R199[5.35]A, R203[5.39]A, and D265[6.58]A mutants showed reduced responses to the wild-type CXCL8[1–65] as compared with the wild-type CXCR1 (Fig. 3d, Supplementary Fig. 7). This result is in agreement with earlier work by Vacchini and colleagues, who tested the effects of natural post-translational modifications of CXCL8[29], and found that either citrullination of R6[CXCL8] (removing its positive charge) or truncation of the N-terminus up to and including this residue increased the potency of the chemokine to inhibit adenylyl cyclase through both CXCR1 and CXCR2. Earlier work by Loos and colleagues also showed that citrullination of CXCL8 increases the ability of the chemokine to mobilise neutrophils[30]. R6[CXCL8] sits between R269[6.62] and R280[7.28], but electrostatic repulsion will be reduced by D265[6.58], whose side-chain is close enough to form hydrogen bonds with both R6[CXCL8] and R269[6.62].

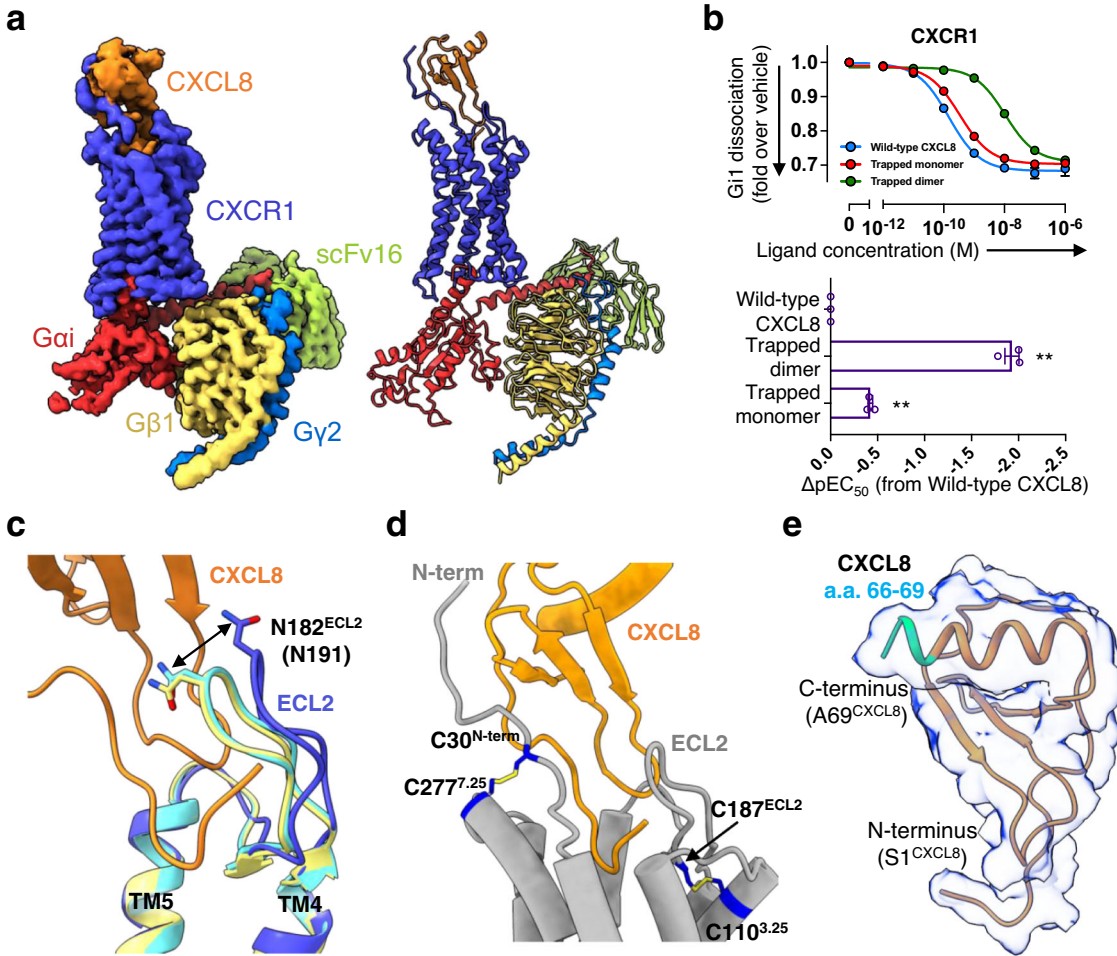

**Fig. 2 | Overall structure and functional analysis of Gi-coupled CXCR1 complex.** **a** The density map of CXCR1-Gi1 heterotrimer-scFv16 complex with the ligand CXCL8. **b** The NanoBiT-based assays used to measure Gi1 dissociation upon treatments with different ligands: the wild-type CXCL8$^{1–72}$ (blue), the trapped monomer (red), and the trapped dimer (green). Symbols and error bars represent mean and SEM, respectively, of three independent experiments. Error bars are not shown where they are smaller than the circle symbols. Bars in the lower panel represent the differences between the calculated potency ($\Delta$pEC$_{50}$) of the trapped dimer and the trapped monomer relative to the wild-type CXCL8$^{1–72}$. For the statistical analyses, ** indicates $P < 0.01$, with one-way ANOVA followed by the Dunnett's test for multiple comparison analysis (with reference to the wild-type CXCL8). **c** The superposition of the structures of CXCR1, CXCR2 (CXCL8$^{1–72}$ bound), and CXCR2 (CXCL8$^{1–65}$ bound), highlighting the differences in the ECL2 region. CXCR1, blue; CXCR2 (CXCL8$^{1–72}$ bound), cyan; CXCR2 (CXCL8$^{1–65}$ bound), lime. **d** The two disulphide bonds at the external face of CXCR1 are shown as stick models. **e** Slice through the density map of CXCL8$^{1–72}$. Residues 66–69, part of the dimeric interface, are clearly visible and marked in green.

CRS1 lies away from the dimer interface of CXCL8, so that CXCR2 bound to the CXCL8 dimer overwhelmingly interacts more with one chemokine subunit than the other. In the structure of the complex formed by CXCR2 and CXCL8 dimer (PDB 6LFM), it is seen that ECL2 of the receptor makes the only contacts with the partner CXCL8 subunit. This structured loop has a notably different sequence in CXCR1 and CXCR2 (Fig. 4a, Supplementary Fig. 4). Comparing models of the two receptors also shows that the N-terminal residue of CXCL8 within CRS2 becomes ordered enough to model the S1$^{CXCL8}$ residue (Figs. 2e, 3a, Supplementary Fig. 6), which is not seen in the CXCR2 complexes. The replacement of alanine in CXCR2 with V186$^{ECL2}$ in CXCR1 appears to be at least partly responsible for this increased ordering of the N-terminus of CXCL8. At the same time, N182$^{ECL2}$, at the tip of the hair-pin in CXCR1, is displaced roughly 4 Å towards the chemokine when compared to the position of the equivalent N191$^{ECL2}$ in CXCR2 (Fig. 2c). This shift would cause clashes with the second subunit of the CXCL8 dimer, mainly through N181$^{ECL2}$, and favour chemokine monomer binding (Fig. 4b). Using the NanoBiT-Gi-dissociation assay, we confirmed that the trapped dimer (CXCL8$^{R26C}$) requires about 85-fold and 32-fold higher concentration than the wild-type and the trapped monomer

(CXCL8$^{1–65}$), respectively, to elicit a response from CXCR1 (Fig. 2b). When ECL2 of CXCR1 was replaced with that of CXCR2, the responses of the chimeric GPCR to the three forms of CXCL8 became very similar to those of CXCR2 (Figs. 2b, 4c, d). Replacing ECL2 of CXCR1 with the equivalent region from CXCR2 confirms that this loop is responsible for the monomer/dimer selectivity of the two receptors.

## Activation of CXCR1

Class A GPCRs share a number of highly conserved motifs, and the conformational changes associated with agonist-induced activation are well known[31]. Agonists trigger significant movements of TM5 and TM6 relative to the inactive state (Supplementary Fig. 5a), and displace a highly conserved "toggle" tryptophan residue (W255$^{6.48}$ in CXCR1) within the conserved CWxP motif of TM6, close to the PIF motif and the NPxxY motif. The DRY motif in TM3 is also common to class A GPCRs, and helps regulate the switch between receptor states. Since CXCR1 and CXCR2 are highly similar, and have all the key motifs in common, the inactive state of CXCR2 can be used with confidence to model the behaviour of CXCR1 (Supplementary Fig. 5a). CXCL8$^{1–72}$ lies considerably more distant (>10 Å) from the toggle tryptophan than is

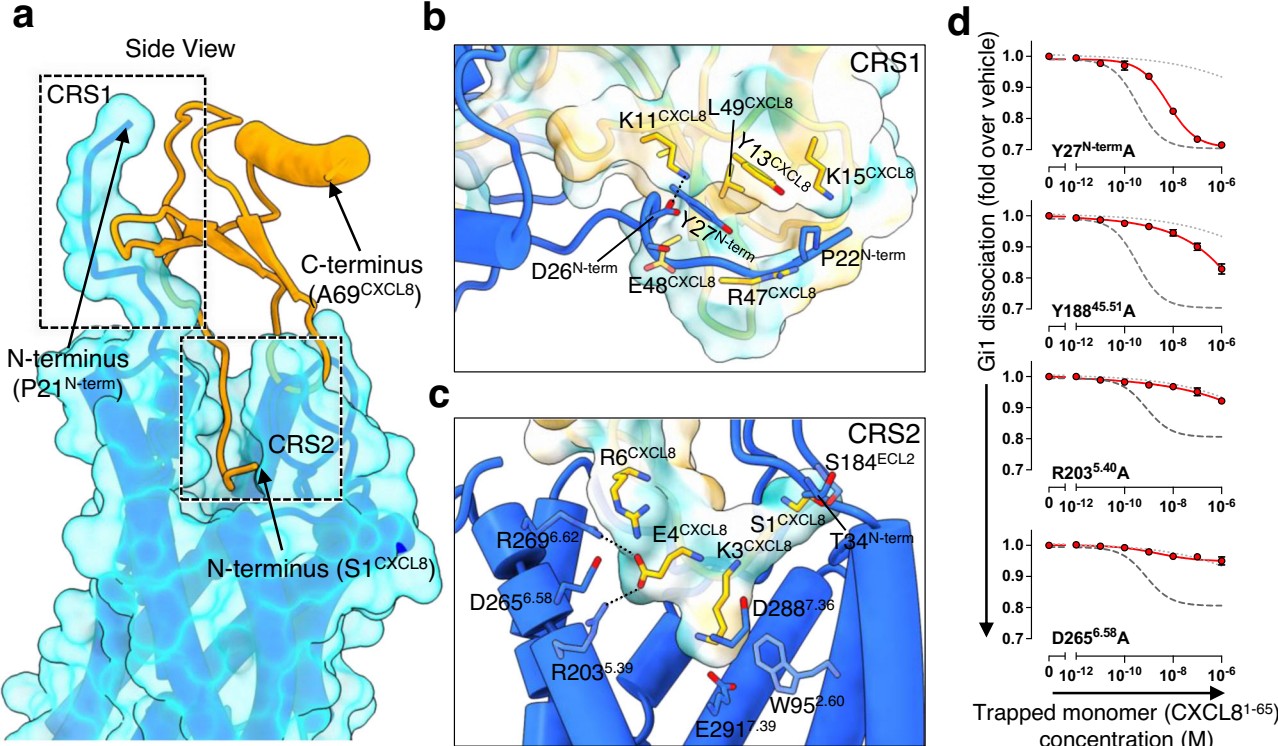

**Fig. 3 | Interaction between CXCR1 and CXCL8. a** View of the interactions between CXCR1 and CXCL8. CRS1 and CRS2 are indicated by dashed boxes. **b** Closer view of CRS1. Side-chains involved in the interaction are shown as stick models. **c** Closer view of CRS2. Hydrogen bonds between CXCR1 and CXCL8 are shown as black dashed lines. **d** The NanoBiT-based assays used to measure Gi1 dissociation for the wild-type CXCR1 (gray dashed lines) and the various mutants (red lines). Non-specific response in the mock-transfected cells is shown as gray dotted lines. Note that the two mutants (R203$^{5.40}$A and D265$^{6.58}$A) showed reduced surface expression and expression-matched WT response is shown (gray dashed lines). Symbols and error bars represent the mean and standard error of three independent experiments.

generally found with small molecule GPCR agonists (Figs. 3a, c, 5a), but it still causes considerable repacking of the interface between helices TM1, TM3, TM5, and TM6. E4$^{CXCL8}$ forms salt bridges with R203$^{5.39}$ and R269$^{6.62}$ in CRS2 (Fig. 2c). These interactions found in CXCR1 and CXCR2 trigger the inward movement of the N-terminal end of TM5, and movement of TM6 towards the cytoplasm (Supplementary Fig. 5a). As TM6 extends towards the cytoplasmic side and rotates, the toggle switch residue W255$^{6.48}$ moves over 3 Å towards I125$^{3.40}$ (Fig. 5b). F251$^{6.44}$ of the PIF motif moves in concert with W255$^{6.48}$ while P214$^{5.50}$ of TM5 and I124$^{3.40}$ of TM3 move towards the ligand pocket (Fig. 5c and supplementary Fig. 5a). In the activated state, Y222$^{5.58}$ forms a hydrogen bond with R135$^{3.50}$ of the DRY motif, allowing Y305$^{7.53}$ of the NPxxY motif to sit in a pocket next to Val 247$^{6.40}$. Conformational changes of the NPxxY and the DRY motifs on CXCR1 activation (Fig. 5d, e) allow the N-terminal end of TM6 to open the binding site for interaction with the G protein C-terminal helix.

### G protein pocket

Interactions between CXCR1 and Gαi are very similar to those seen in other GPCR complexes. The C-terminal residues of the G protein such as I344$^{G.H5.16}$, C351$^{G.H5.23}$, L353$^{G.H5.25}$, F354$^{G.H5.26}$ make apolar contacts with the N-terminus of TM6 and the C-terminal end of TM3 (Fig. 5f). CXCR1 and CXCR2 are highly similar in this region, making significant differences unlikely in their interactions with the G protein. The only residue differing in CXCR1 from CXCR2 in this G-protein contact area is N311$^{8.49}$, whose equivalent residue is K320$^{8.49}$ in CXCR2 (Supplementary Fig. 4), which may exert some repulsion on K349$^{G.H5.21}$ of Gαi. Minor contacts are also seen between ICL2 and Gαi, principally through R144$^{34.52}$ and R150$^{34.58}$ (Fig. 5g). This loop has the same sequence in

CXCR1 and CXCR2, although the details of its interaction differ in our model and published CXCR2 models[27].

## Discussion

Chemokines are a very ancient family of proteins that may have arisen before jawed animals[32]. They occupy a central position in inflammatory responses and other processes which require fine control. Numerous cell-surface receptors have therefore evolved to produce the appropriate response in target cells, so that a chemokine may elicit different behaviours in different cell types. Chemokines form largely rigid structures stabilized by disulphide bridges, but may undergo a variety of post-translational modifications. Here we have used bacterial expression to produce different forms of CXCL8, and used mass spectrometry to confirm the nature and uniformity of our samples. Highly charged ions with a broad distribution are generally observed in the mass spectra of the acid-denatured proteins, but both trapped dimer (CXCL8$^{R26C}$) and trapped monomer (CXCL8$^{1-65}$) presented weakly charged ions with a narrow charge distribution, similar to those observed under non-denaturing conditions. The chemokine samples are therefore natively folded, and carry an intact N-terminal tail. This flexible region forms interactions with a surface pocket on the receptors, whose own flexible N-terminal tail binds to a surface groove on the chemokine. CXCL8 binding triggers calcium release and chemotaxis through both CXCR1 and CXCR2[33], but only CXCR1 mediates phospholipase D activation and respiratory burst[34]. These functional differences are mirrored by the preference of CXCR1 for the monomeric CXCL8, although no underlying connection is yet apparent. Although the native chemokine associates weakly, at higher concentration CXCL8 will compete with itself for binding to CXCR1,

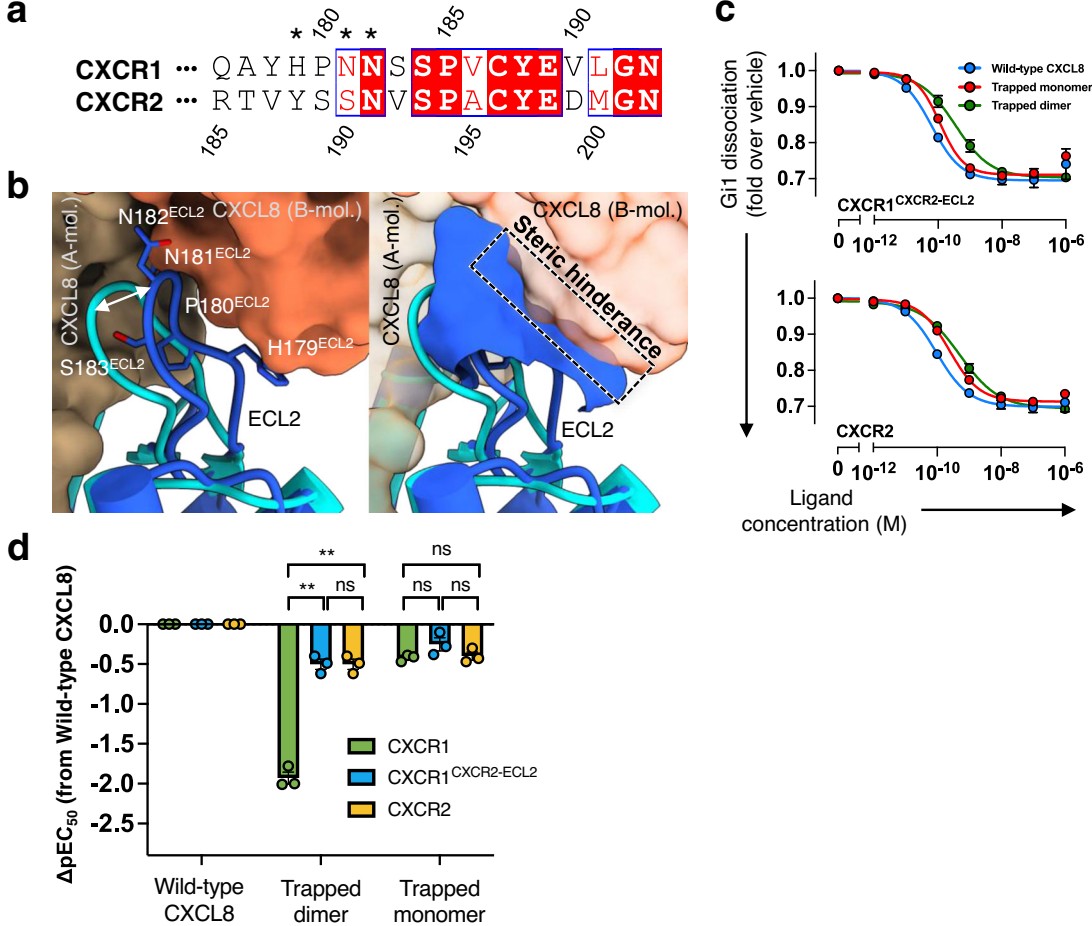

**Fig. 4 | Ligand selectivity of CXCR1. a** Sequence alignment of the residues forming ECL2 in CXCR1 and CXCR2. Residues with asterisks cause steric clashes when the dimer state of CXCL8 is docked onto the protein. **b** The superposition of CXCR1 (blue) onto CXCR2 (cyan) with CXCL8$^{1-72}$ dimer bound. (left panel) Two CXCL8 molecules are shown as a surface model, while the residues that form ECL2 are represented as a stick models. (right panel) ECL2 of CXCR1 is shown as a surface model, and the region of steric clash with dimeric CXCL8 is indicated by a box (dashed black). **c** The NanoBiT-based assays used to measure Gi1 dissociation with the wild-type CXCR2, and the chimeric CXCR1 carrying the ECL2 of CXCR2 (the wild-type CXCL8$^{1-72}$, blue; the trapped monomer, red; the trapped dimer, green). **d** Bars represent the $\Delta pEC_{50}$ of the trapped dimer and the trapped monomer relative to the wild-type CXCL8$^{1-72}$ for the wild-type CXCR1, CXCR2, and the CXCR1-ECL2 chimera (CXCR1$^{CXCR2-ECL2}$). Symbols and error bars represent mean and SEM, respectively, of three independent experiments with each performed in duplicate. For the statistical analyses, ** indicates $P < 0.01$, with two-way ANOVA followed by the Dunnett's test for multiple comparison analysis (with reference to the wild-type CXCR1). ns, not significantly different between the groups.

limiting its agonist effect, and it may be speculated that this allows gradient sensing.

The sequence differences between CXCR1 and CXCR2 are largely clustered into three segments: the N-terminal region, the second extracellular loop, ECL2, and the C-terminal cytoplasmic tail. Numerous studies have been undertaken to map the functional differences between the two receptors, and their interactions with chemokines, to these regions. GROα and NAP-2 show no effect with native CXCR1, but replacing the N-terminal peptide of the receptor with the sequence from CXCR2 allowed the chimeric receptor to respond to these chemokines as well as CXCL8[35]. Granulocyte chemotactic protein 2 (GCP-2), better known as CXCL6, is a 75 residue protein with 31% sequence identity to CXCL8. It induces chemotaxis and calcium release in neutrophils through interactions with CXCR1 and CXCR2, which it binds with nanomolar affinity, but its role is less studied than CXCL8[36,37]. CXCL6 triggers responses as effectively as CXCL8 does through CXCR2, but less so through CXCR1; replacing R20 of CXCL6 (equivalent to K15$^{CXCL8}$ of CXCL8) with glycine has no effect on binding to CXCR2, but renders the chemokine ineffective against CXCR1[37]. Correlating such functional effects with the structural models is not trivial, and there are no obvious interactions made by K15$^{CXCL8}$ of CXCL8 for

example to suggest its importance. K11$^{CXCL8}$ of CXCL8 on the other hand sits close to D26$^{N-term}$ of CXCR1. High-affinity binding alone however does not imply a highly effective agonist.

In conclusion, CXCR1 and CXCR2 share common features of the activation mechanism found for other class A GPCRs. Our refined model reveals details of the interactions between CXCR1 and CXCL8, and shows that steric clashes with ELC2 are responsible for the known preference of the receptor for monomeric rather than dimeric chemokine. Although both CXCR1 and CXCR2 form similar agonist contacts in the CRS2 region, they have different N-terminal and ECL2 sequences and make substantially different interactions with CXCL8 at the CRS1 interface, which is largely responsible for chemokine selectivity.

## Methods

### Expression and purification of CXCR1 and CXCL8 proteins

DNA encoding human CXCR1 (UniProtKB-P25024), codon-optimized for expression in *Spodoptera frugiperda* (Sf9) cells, was synthesized by Genscript. The gene was cloned into a modified pFastBac HT-B vector containing N-terminal haemagglutinin (HA) signal sequence, Flag-tag (DYKDDDD), thermostabilized apocytochrome b562RIL (BRIL), HRC3V

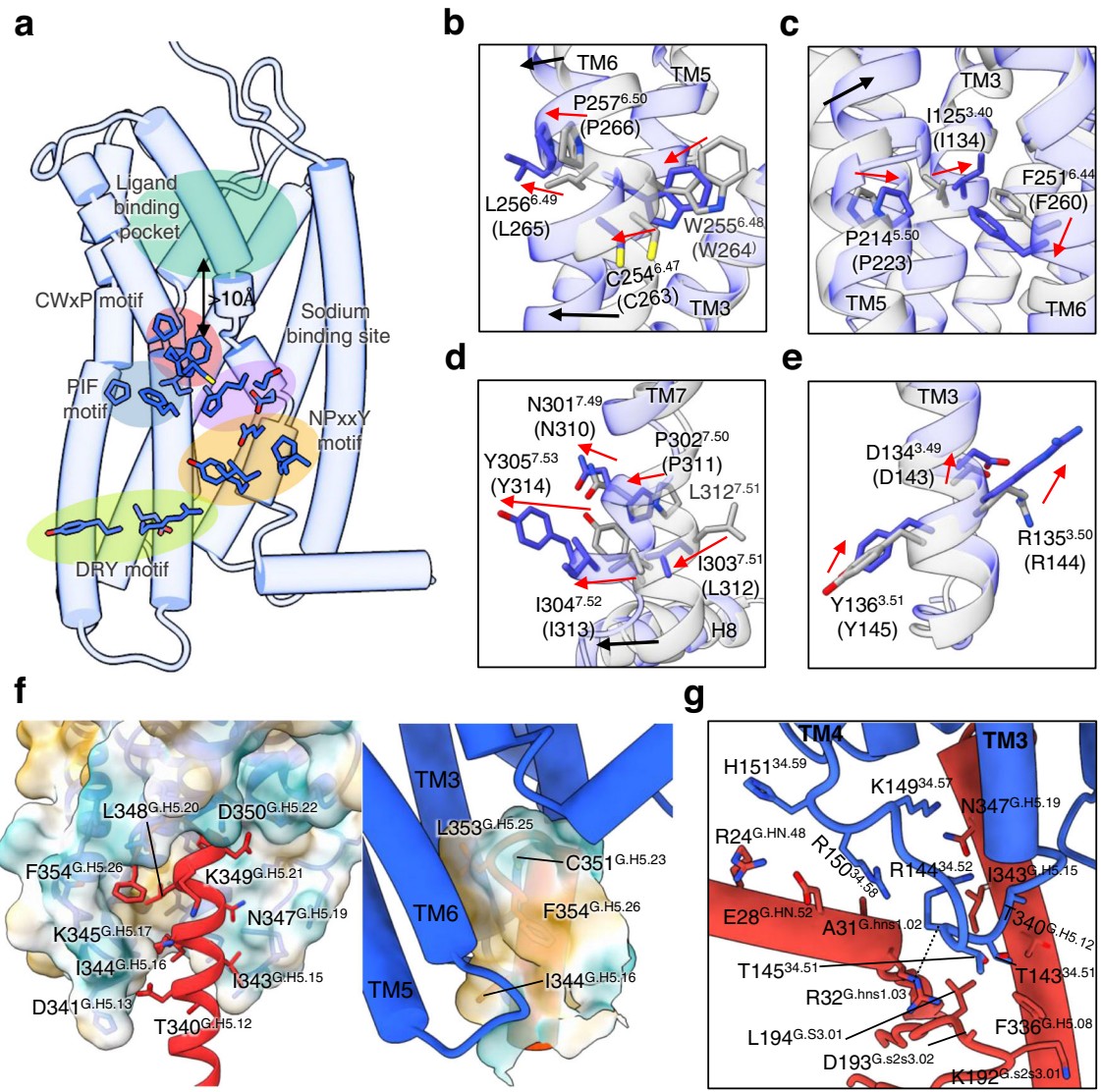

**Fig. 5 | Activation mechanism of CXCR1 and Gαi protein interactions.** Motifs within CXCR1, shown as coloured patches (green, ligand binding site; red, CWxP motif; blue, PIF motif; purple, sodium site; orange, NPxxY motif; yellow-green, DRY motif). **b**–**e** Magnified views of these motifs within the CXCR1 complex (blue) and the antagonist-bound inactive CXCR1 (grey). Each residue is shown as a stick model. Residue names in parentheses are equivalent residues from CXCR2. The conformational changes are shown around (**b**) the CWxP motif, (**c**) the PIF motif, (**d**) the NPxxY motif and (**e**) the DRY motif. **f** Cartoon representation of the CXCR1 complex illustrating the receptor-Gαi interfaces. CXCR1 and the helix 5 of Gαi are shown as molecular surface models in the left and right panels, respectively. **g** The interaction between ICL2 of CXCR1 and Gαi. Helices are shown as cylinders. Residues involved contacts between the proteins are shown as stick models.

protease cleavage site and C-terminal 8 × His tag. CXCR1 was expressed in Sf9 insect cells using a Bac-to-bac expression system. Cells were infected at a density of $4 \times 10^6$ cells per ml and incubated at 27 °C for 60–72 h until harvest. To purify membrane fractions, the harvested cells were first disrupted in a hypotonic buffer containing 10 mM HEPES pH 7.5, 10 mM MgCl₂, and 20 mM KCl, and then in a high osmotic buffer containing 10 mM HEPES pH 7.5, 1 M NaCl, 10 mM MgCl₂, and 20 mM KCl in the presence of a protease inhibitor cocktail (Roche). Purified membranes were solubilized at 4 °C for 2 h in a buffer containing 50 mM HEPES pH 7.5, 500 mM NaCl, 10% glycerol, 1% (w/v) n-dodecyl-beta-D-maltopyranoside (DDM, Anatrace), 0.2% (w/v) cholesterol hemisuccinate (CHS, Anatrace) with 2 µM CXCL8$^{1-72}$. Insoluble debris was removed by ultracentrifugation in 264,902 g at 4 °C for 1 h, and the supernatant was incubated with TALON® metal affinity resin (Clontech) overnight at 4°C. The resin was washed three times with 10 column volumes of washing buffer I containing 25 mM HEPES pH 7.5, 500 mM NaCl, 10% glycerol, 20 mM IMD, 0.05% DDM, 0.01% CHS, 2 µM CXCL8$^{1-72}$ and then washing buffer II containing 25 mM HEPES pH 7.5,

500 mM NaCl, 10% glycerol, 20 mM IMD, 0.05% LMNG, 0.005% CHS, 0.01% DDM, 0.002% CHS, 2 µM CXCL8$^{1-72}$ and then another 10 column volumes of washing buffer III containing 25 mM HEPES pH 7.5, 500 mM NaCl, 10% glycerol, 20 mM IMD, 0.05% LMNG, 0.005% CHS, 2 µM CXCL8$^{1-72}$. The protein was then eluted using the washing buffer III supplemented with 350 mM imidazole. Eluted protein was concentrated and passed through a PD-10 Desalting Column (Cytiva). HRV3C protease was added to cleave the Flag-tag and BRIL, and removed by nickel affinity chromatography. Fractions containing CXCR1 were pooled and subjected to size-exclusion chromatography on a Superdex 200 Increase 10/300 GL column (Cytiva) pre-equilibrated in 20 mM HEPES pH 7.5, 100 mM NaCl, 1 mM MgCl₂, 0.5 mM TCEP, 0.05% LMNG, 0.005% CHS in presence 2 µM CXCL8$^{1-72}$. The fractions containing CXCR1 were collected and concentrated before analysis.

DNA encoding human CXCL8 (UniProtKB-P10145) was synthesized by Genscript. The gene was inserted into the cloning site of pET-32a vector, such that the expressed protein carries an N-terminal

sequence including a histidine tag followed by a TEV protease cleavage site (ENLYFQ|S). After TEV treatment, the expressed chemokine has the N-terminal sequence SAKELR. The plasmid was transformed into *E. coli* BL21 (DE3) cells. The transformed cells were grown in LB medium at 37 °C. When the optical density at 600 nm reached 0.7, IPTG was added to 0.5 mM final concentration.

Harvested cells were then lysed by sonication in a buffer containing 20 mM HEPES pH 7.0, 150 mM NaCl, 0.2% Triton-X100, 0.25 μM PMSF. After centrifugation at 4 °C for 30 min, the supernatant was collected, and the CXCL8 was purified by Ni-NTA affinity chromatography. The eluate was treated with TEV protease, and the tag-free CXCL8 protein was applied to an anion exchange chromatography and Superdex75 increase 10/300 GL column (Cytiva), equilibrated with a buffer containing 20 mM HEPES pH 7.5, 150 mM NaCl.

### Expression and purification of heterotrimeric G-protein and scFv16

The same construct was used for heterotrimeric Gαi1, Gβ1 and Gγ2 expression as a previous report[38]. The scFv16 gene was cloned into a modified pFastBac1 vector with N-terminal GP67 signal sequence and C-terminal TEV protease cleavage site followed by a 6 histidine-tag. Purification was carried out as before[38] with simple modification. In brief, media containing secreted scFv16 was separated by centrifugation at 72-96 hours after infection. The pH of the medium was adjusted to 7.5-8.0 by adding Tris-base powder. Chelating agents were quenched by addition of 1 mM nickel chloride and 5 mM calcium chloride and stirring at room temperature for 1 h. After centrifugation, the supernatant was mixed and incubated with 5 ml of Ni-EXCEL resin (Cytiva). After 2 hours, the resin was collected and washed with 20 column volumes of buffer containing 20 mM HEPES pH 7.5, 500 mM NaCl and 20 mM imidazole. The scFv16 was eluted with 20 mM HEPES pH 7.5, 100 mM NaCl and 350 mM imidazole. After TEV protease treatment, the sample was further purified by size-exclusion chromatography using a Superdex 200 16/600 pg column (Cytiva). The peak fraction was collected and concentrated to 5 mg/mL for future use.

### Formation of CXCR1–Gi1 heterotrimer-scFv16 complex

CXCR1 and Gαi1 heterotrimer were mixed in a 1:1.2 molar ratio with 2.5 μl of Apyrase (NEB) in presence 2 μM CXCL8$^{1-72}$ and incubated at 25 °C for 30 min. Purified scFv16 was added to a 1:1.3, Gi heterotrimer:scFv16 molar ratio. The mixture was incubated on ice for 1 h. Then the mixture was subjected to size-exclusion chromatography on a Superdex 200 Increase 10/300 GL column (Cytiva) pre-equilibrated with 20 mM HEPES pH 7.5, 100 mM NaCl, 1 mM MgCl$_2$, 0.5 mM TCEP, 0.001% LMNG, 0.0001% CHS. Peak fractions containing CXCR1/Gi/CXCL8$^{1-72}$ complex were collected and concentrated to 2.5 mg/mL.

### Cryo-EM grid preparation and data collection

3 μL of the sample was applied to a glow-discharged holey carbon grid (Quantifoil R1.2/1.3, Cu, 300 mesh). The grid was blotted for 5 s at 4 °C, 100% humidity and flash-frozen in liquid ethane using Vitrobot Mark IV instrument (Thermo Fisher Scientific) before storage in liquid nitrogen. Cryo-EM images were collected using a 300 kV Titan Krios G4 (Thermo Fishier Scientific) in Riken, Yokohama, Japan, equipped with a K3-summit camera with 15 eV slit. 4,175 movies were collected by standard mode for 48 frames with a total dose of 51.16 e/Å2, exposure time of 4.7 s, and dose on camera at 7.5 e$^{-1}$/px/sec. Magnification of micrographs was ×105,000, and pixel size was 0.83 Å/pixel. Defocus range was −0.8 to −1.6 μm. Data collection was automated using EPU software.

### Cryo-EM data processing

The collected data were processed by cryoSPARC (v.3.3.1)[39], beginning with Patch motion correction and CTF estimation. Micrographs under 5 Å CTF resolution were selected using Curate Exposures, giving 4,065 micrographs for analysis. Particles were auto-picked with blob picker, using all micrographs. 2,046,429 particles were extracted with binning state 3.35 Å/pix. Suitable particle classes from 2D classification were used to make 3D models ab initio. A 3D model of the entire protein complex was then used to re-run 2D classification. Using selected suitable 2D particle classes as reference, particles were auto-picked by Template picker from all micrographs. 2,530,947 particles were picked from micrographs with binning state 3.35 Å/pix. After further 2D classification, particles were selected and classified by Ab-Initio Reconstruction into 4 classes. Hetero-refinement was performed three times. One class showed the complete complex, and particles of this class were extracted with high-resolution (1.09 Å/pix). Subsequently Relion (v.4.0.0)[40] was used for 2D classification of these particles, giving 282,687 in all. After multibody refinement focusing on the cytokine ligand, a new mask was made to cover the complete complex. 3D classification was performed with Relion, a class was chosen with strong ligand density. Finally, 120,631 particles were selected and 3D reconstruction performed with non-uniform refinement by cryoSPARC, and a 3.41 Å map was obtained. The final map for modelling was calculated by deepEMhancer (v.0.13)[41].

### Model building and refinement

The model of CXCR1 was used from the AlphaFold Protein Structure Database (UniProt: P25024). Models of Gi heterotrimer and scFv16 complexes were taken from the structure of CXCR2/CXCL8(monomer) complex (PDB 6LFO). Models and cryo-EM map were roughly fitted by ChimeraX[42] and Real-Space Refine in PHENIX[43]. The model was built manually based on Cα and side-chain maps using COOT[44] and further refined using Real-Space Refine in the PHENIX suite.

### Electrospray ionisation mass spectrometry

ESI mass spectra were obtained with a SYNAPT G2 HDMS mass spectrometer equipped with a nanoelectrospray (nanoESI) ion source (Waters, Milford). Prior to nanoESI-MS, the protein solutions (20 mM HEPES, 100 mM NaCl) were replaced with 200 mM ammonium acetate by gel filtration with Micro Bio-Spin 6 (BioRad, Hercules) or dialysis. For obtaining the mass spectra under non-denaturing conditions, 5–20 μM protein samples in 200 mM ammonium acetate were subjected to nanoESI-MS. For obtaining the mass spectra under denaturing conditions, the protein samples in 200 mM ammonium acetate were mixed with formic acid and methanol, resulting in 2–3.5 μM proteins in 100 mM ammonium acetate containing 2–3% formic acid and 30–40% methanol. The pH of the protein solutions under the denaturing conditions was confirmed by pH test paper to be pH-1. A few microliters of the sample solution were deposited in a nanoESI emitter (HUMANIX, Japan). The parameters used for the measurement were as follows: ion source temperature 70 °C; capillary voltage 0.7–0.85 kV; sampling cone voltage 25–40 V. Spectra were obtained by acquiring the data for 2 min in the mass range of m/z 500–4000 (denaturing conditions) or 1000–4000 (non-denaturing conditions). Mass spectra were processed using MassLynx 4.2 (Waters).

### NanoBiT-G-protein dissociation assay

CXCR1/2-induced Gi1 activation was measured by the NanoBiT-G-protein dissociation assay[45], in which the interaction between a Gα subunit and a Gβγ subunit was monitored by the NanoBiT system (Promega). Specifically, a NanoBiT-Gi1 protein consisting of Gαi1 subunit fused with a large fragment (LgBiT) at the α-helical domain (between the residues 91 and 92 of Gαi1) and an N-terminally small fragment (SmBiT)-fused Gγ2 subunit with a C68S mutation was expressed along with untagged Gβ1 subunit and CXCR1 or CXCR2 (containing the N-terminal HA-derived signal sequence followed by the FLAG-epitope tag). HEK293A cells were seeded in a 6-cm culture dish at a concentration of $2 \times 10^5$ cells ml$^{-1}$ (4 ml per well in DMEM (Nissui) supplemented with 10% fetal bovine serum (Gibco), glutamine,

penicillin, and streptomycin), one day before transfection. Transfection solution was prepared by combining 10 μL (per dish hereafter) of polyethylenimine (PEI) Max solution (1 mg ml$^{-1}$; Polysciences), 400 μL of Opti-MEM (Thermo Fisher Scientific) and a plasmid mixture consisting of 400 ng CXCR1 or CXCR2 (or an empty plasmid for mock transfection), 200 ng LgBiT-containing Gαi1 subunit, 1 μg Gβ1 subunit and 1 μg SmBiT-fused Gγ2 subunit (C68S). For the lowered expression of the wild-type CXCR2, 40 ng plasmid (10-fold less) of CXCR2 plasmid was used. After incubation for 1 day, the transfected cells were harvested with 0.5 mM EDTA-containing Dulbecco's PBS, centrifuged, and suspended in 2 ml of HBSS containing 0.01% bovine serum albumin (BSA; fatty acid-free grade; SERVA) and 5 mM HEPES (pH 7.4) (assay buffer). The cell suspension was dispensed in a white 96-well plate at a volume of 80 μL per well and loaded with 20 μL of 50 μM coelenterazine (Angene) diluted in the assay buffer. After a 2 h incubation at room temperature, the plate was measured for baseline luminescence (SpectraMax L, Molecular Devices) and titrated concentrations of niacin (20 μL; 6X of final concentrations) were manually added. The plate was immediately read at room temperature for the following 5 min, at measurement intervals of 20 s. The luminescence counts from 5 min to 10 min after ligand addition were averaged and normalized to the initial count. The fold-change values were further normalized to those of vesicle-treated samples and used to plot the G-protein dissociation response. Using Prism 9 (GraphPad Prism), the G-protein dissociation signals were fitted to a four-parameter sigmoidal concentration-response curve with a constraint of the *HillSlope* to absolute values less than 2. For each replicate experiment, *Top* and *Bottom* parameters of the wild-type, the trapped dimer and the trapped monomer of CXCL8 were constrained to be "shared values". Thereafter, the parameters *Span* (= *Top* − *Bottom*) and $pEC_{50}$ (negative logarithmic values of $EC_{50}$ values) of individual receptor constructs were normalized to those of the wild-type CXCL8 performed in parallel and the resulting $\Delta pEC_{50}$ values were used to calculate ligand response activity of the ligands.

### Flow cytometry

Transfection was performed according to the same procedure as described in the 'NanoBiT-G-protein dissociation assay' section. After incubation for 1 day, the transfected cells were detached and then dispensed in a 96-well V-bottom plate and fluorescently labeled by using an anti-FLAG epitope (DYKDDDDK) tag monoclonal antibody (Clone 1E6, FujiFilm Wako Pure Chemicals; 10 μg ml$^{-1}$) and a goat anti-mouse IgG secondary antibody conjugated with Alexa Fluor 488 (Thermo Fisher Scientific; 10 μg ml$^{-1}$), as described previously in ref. 45. The resulting cells were analyzed by an EC800 flow cytometer equipped with a 488 nm laser (Sony). The fluorescent signals using all of the recorded events were analyzed with FlowJo software (FlowJo). Live cells were gated with a forward scatter (FS-Peak-Lin) cutoff at a value of 390 with a gain value of 1.7. Mean fluorescence intensity from all of the recorded events (approximately 20,000 cells per sample) was analyzed by a FlowJo software (FlowJo) and used for statistical analysis. For each experiment, we normalized an MFI value of the mutants by that of the wild-type performed in parallel and denoted relative levels.

### Reporting summary

Further information on research design is available in the Nature Portfolio Reporting Summary linked to this article.

## Data availability

The data that support this study are available from the corresponding authors upon request. The cryo-EM map has been deposited in the Electron Microscopy Data Bank (EMDB) under accession code EMD-35351 (CXCR1/Gi/CXCL8$^{1-72}$). The atomic model has been deposited in the Protein Data Bank (PDB) under accession code 8IC0 (CXCR1/Gi/

CXCL8$^{1-72}$). Previously published structures referenced can be found under accession codes 6LFL and 6LFO. Source data for the results of the NanoBiT-G-protein dissociation assay and adjusted *P* values are provided as a Source Data File. Source data are provided with this paper.

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

## Acknowledgements

The cryo-EM experiments were performed at the cryo-EM facility of the RIKEN Center for Biosystems Dynamics Research (Yokohama). We would like to thank Kayo Sato, Shigeko Nakano and Ayumi Inoue at Tohoku University for their assistance in plasmid preparation and the cell-based GPCR assays. This work was supported by Japan Agency for Medical Research and Development (AMED) under grant numbers JP21fk0310103 (S-Y.P.), JP20gm0010004 (A.I.), JP22zf0127007 (A.I.), JP22ama121038 (A.I.) and JP20am0101095 (A.I.); JSPS/MEXT under KAKENHI grant numbers JP19H05779 (S-Y.P.), JP21H02449 (S-Y.P.), JP19H05774 (S.A.), JP21H04791 (A.I.), JP21K19236 (S.A.), JP21H05113 (A.I.), JPJSBP120213501 (A.I.) and JPJSBP120218801 (A.I.); Japan Science and Technology Agency (JST) grants JPMJFR215T (A.I.) and JPMJMS2023 (A.I.). This work was supported in part by JST, the establishment of university fellowships towards the creation of science technology innovation, grant number JPMJFS2140 (N.I.).

## Author contributions

N.I., J-H.P., and K.M. expressed and purified proteins. N.I., J-H.P., J.R.H.T. and S-Y.P. collected and analyzed the cryo-EM data and refined the structure. J-H.P., N.I., S-Y.P., K.K., and A.I. designed the mutants. K.K., and A.I. performed the cell-based mutant assays. M.T and S.A. performed and analyzed the ESI-MS. S-Y.P. initiated the project, planned and supervised the research. N.I., J-H.P., J.R.H.T and S-Y.P. wrote the manuscript with input from all co-authors.

## Competing interests

The authors declare no competing interests.
