## [Peer Review File · Nature Communications]

Structural basis of CXC chemokine receptor 1 ligand binding and activationReviewers' Comments:

Reviewer #1:

Remarks to the Author:

This manuscript from Ishimoto etc. reports a cryo-EM structure of the CXCR1-Gi complex with CXCL8 (IL-8), which provides detailed molecular insights into the molecular interactions between the receptor and the chemokine and Gi. The authors also discuss the selectivity mechanism of CXCR1 for monomeric over dimeric IL-8. Overall, this study provides a comprehensive and high-quality analysis of the structure of a significant chemokine and its receptor. The conclusions drawn from the structural and functional data are well-supported, and the figures presented are clear and informative.

Some minor issues:

1. IL-8 was produced from E.coli, which lacks proper post-translational modulations. Will it affect the dimerization of IL-8?
2. The resolution of several regions of IL-8 in the cryo-EM structure is below 5 angstrom, which limits the confidence with which side chains can be accurately modeled. This limitation should be acknowledged and discussed in the main body of the manuscript to ensure transparency and scientific rigor.
3. It would be valuable to mention whether ICL2 of CXCR1 interacts with Gi. Additionally, providing a structural comparison between the CXCR1-Gi and CXCR2-Gi complexes would elucidate the similarity of these two receptors.

Reviewer #2:

Remarks to the Author:

See attached file.

Overview

Chemokine activation of chemokine receptors is responsible for directing migration of leukocytes in inflammation. In particular, CXCR1 and CXCR2 are the main chemokine receptors on neutrophils, but they respond to different (although overlapping) groups of chemokines. Moreover, they have different selectivity for monomeric and dimeric forms of their key shared chemokine ligand, interleukin 8 (IL8 or CXCL8), which may have implications for how they sense chemokine gradients.

This manuscript describes the cryoEM structure of CXCR1 bound to IL8 (bound as a monomer) and to a heterotrimeric G protein (with a stabilising antibody fragment). This represents the active state of the receptor. Previous structures have been reported for CXCR2 in the inactive state, the active state bound to monomeric IL8, and the active state bound to dimeric IL8.

Major Points

1. Considering that the IL8-CXCR1 interaction is, in many ways, the archetype of chemokine-receptor interactions, having the structure of this complex in the database is of substantial value.
2. Most features of the new structure are very similar to the active state structures of CXCR2, so there are not really any new insights about the mechanism of receptor activation.
3. On the other hand, in the region where IL8 dimerises, there is a significant difference between the structures of CXCR1 and CXCR2 (in the ECL2 region), which accounts for the observation that CXCR2 responds similarly to trapped monomeric and dimeric forms of IL8, whereas CXCR1 has about a 30-fold preference for the monomeric form. This is shown convincingly using a chimeric receptor in which the ECL2 region of CXCR1 is replaced by that of CXCR2 (Fig 4c,d). This was an elegant experiment and is probably the highlight of the paper.
4. The authors also point out that there are substantial differences in the N-terminal sequences of the two receptors, which bind to the chemokine (in chemokine receptor site 1; CRS1) and are suggested to be responsible for the different chemokine selectivity of the receptors. However, no experiments are presented to test the importance of specific residues for chemokine selectivity.
5. The Discussion does not, in my view, make a strong case for the importance of this new study. In fact, it says very little about the key finding. What is the importance of understanding the structural basis by which CXCR1 and CXCR2 respond differently to monomeric and dimeric IL8? For example, does it have implications for how neutrophils sense chemokine gradients?
6. The second paragraph of the Discussion could, potentially, be better developed to explain how the new structure sheds light on the chemokine selectivity of these receptors and why this is of biological importance. For example, does it suggest any strategies for selectively blocking the responses to certain chemokines? Lines 293-303 focus on differences between CXCL6 and IL8, but didn't come to any clear conclusion (or maybe I missed the point).
7. Discussion lines 280-283. It is interesting that CXCR1, but not CXCR2, mediates phospholipase D activation. The authors suggest that this is related to the preference of CXCR1 for monomeric IL8. I don't see the connection. Does the trapped monomer mediate phospholipase D activation via both receptors? What are the other possible explanations for only CXCR1 stimulating the phospholipase D response?

Minor Points

1. Systematic nomenclature uses CXCL8 instead of IL8. I suggest this should be changed throughout the manuscript.
2. Line 44 “dimeric” should say “monomeric”
3. Line 57 CX₃C (3 should be a subscript)
4. Line 149: 1a should be 2a
5. Fig 5b. The sodium site (light blue) is very difficult to see (I’m not sure it’s actually shown).
6. Line 207: Should this be CRS2, not CSR1?
7. Line 227: CXCR2, not CXCLR2
8. Discussion lines 273-277. This comment on the mass spectra seems out of place here.
9. Line 349. Please indicate what protein construct is encoded after cloning the IL8 gene into pET-32a. I assume there is some N-terminal fusion that is removed after TEV cleavage.

We thank the reviewers for their comments, which have been very helpful in improving the manuscript. Changes to the text are highlighted in red in the revised version. Our responses to the referees' comments are detailed below.

We would greatly appreciate your consideration of this manuscript for publication.

Reviewer 1.

- 1) *IL-8 was produced from E.coli, which lacks proper post-translational modulations. Will it affect the dimerization of IL-8?*

There are two main post-translational modifications of IL-8, which are citrullination of Arg6 or removal of the first 6 residues. The truncated form is the major natural isoform. Analyses of the effects of these modifications has been published, but these do not address the dimerization of the cytokine. Arg6 and the preceding residues are not at the dimer interface of IL-8, and are not expected to affect the dimerization. Extra citations and text have now been included to clarify these points.

- 2) *The resolution of several regions of IL-8 in the cryo-EM structure is below 5 angstrom, which limits the confidence with which side chains can be accurately modeled. This limitation should be acknowledged and discussed in the main body of the manuscript to ensure transparency and scientific rigor.*

The nominal resolution of the map is 3.4 Angstrom, which is substantially higher resolution than 5 Angstrom. Additional maps are now shown to give the reader a better impression of the accuracy of side-chain placement (Supplementary Fig. 3b), and comments have been added in the main text to warn the reader against assuming accuracy of the model.

- 3) *It would be valuable to mention whether ICL2 of CXCR1 interacts with Gi. Additionally, providing a structural comparison between the CXCR1-Gi and CXCR2-Gi complexes would elucidate the similarity of these two receptors.*

A comment has been added in the text about ICL2. This loop is conserved between CXCR1 and CXCR2. Extra figures (Fig. 5g and Supplementary Fig. 5c) have also been added to the paper.

Reviewer 2.

Major Points;

The reviewer makes a number of points, several of which require no modification of the paper, so we focus our reply on those comments calling for changes. We are however pleased that the reviewer thinks our structure is of substantial value, and that the external loop exchange experiment was elegant.

- 4) *The authors also point out that there are substantial differences in the N-terminal sequences of the two receptors, which bind to the chemokine (in chemokine receptor site 1; CRS1) and are suggested to be responsible for the different chemokine selectivity of the receptors. However, no experiments are presented to test the importance of specific residues for chemokine selectivity.*

With regard to (4), the reviewer suggests that no experiments are presented to test specific residues for a role in chemokine selectivity. We have in fact made and tested a number of mutants on the basis of our structural work (as shown in Fig. 3 and Supplementary Fig. 7). Our manuscript refers to earlier work, for example studies carried out with rabbit IL-8, and shows how the much weaker binding of the animal protein fits our model. We have now included extra sentences (noted above) showing that our model also clearly explains the increased potency of IL-8 truncated at the N-terminus, or with a citrullinated arginine at the N-terminus. Two other papers are now cited in the Introduction which showed the important role of the receptor N-terminus in determining chemokine specificity and cellular response.

- 5) *The Discussion does not, in my view, make a strong case for the importance of this new study. In fact, it says very little about the key finding. What is the importance of understanding the structural basis by which CXCR1 and CXCR2 respond differently to monomeric and dimeric IL8? For example, does it have implications for how neutrophils sense chemokine gradients?*

In point (5) the reviewer suggests the Discussion does not make a strong case for the importance of the study, and says very little about the key finding. However, a sentence in the final paragraph states: Our refined model reveals details of the interactions between CXCR1 and IL-8, and shows that steric clashes with ELC2 are responsible for the known preference of the receptor for monomeric rather than dimeric chemokine.

This sentence seems to us to summarize our key finding succinctly. Perhaps the reviewer finds the Discussion unduly modest, but we see no reason to elaborate it greatly, especially in the light of Reviewer 1's caution not to overinterpret our data.

- 6) *The second paragraph of the Discussion could, potentially, be better developed to explain how the new structure sheds light on the chemokine selectivity of these receptors and why this is of biological importance. For example, does it suggest any strategies for selectively blocking the responses to certain chemokines? Lines 293-303 focus on differences between CXCL6 and IL8, but didn't come to any clear conclusion (or maybe I missed the point).*

In point (6) the reviewer suggests that more could have been discussed regarding chemokine selectivity, and asks whether our model suggests strategies for selectively blocking responses to certain cytokines. In fact we mention that CXCR1 is less responsive to IL-6 than CXCR2, and that mutations of IL-6 may have very different effects on the responses elicited by the two receptors. We also note that correlating such functional effects with the structural models is not trivial. CXCR1 and CXCR2 have very different,

flexible N termini, and cytokine selectivity can be switched by swapping this region (as noted in the paper). We can speculate of course, but we have chosen to limit our analysis to the new data presented. It seems unlikely that cytokines such as IL-6 bind in locations far removed from the binding site for IL-8, although it is possible that allosteric drugs may be developed with different effects on stimulation by different cytokines. Our paper suggests no quick route to the discovery of such molecules.

7) *Discussion lines 280-283. It is interesting that CXCR1, but not CXCR2, mediates phospholipase D activation. The authors suggest that this is related to the preference of CXCR1 for monomeric IL8. I don't see the connection. Does the trapped monomer mediate phospholipase D activation via both receptors? What are the other possible explanations for only CXCR1 stimulating the phospholipase D response?*

In point (7) the referee claims that we have suggested that phospholipase activation by CXCR1 is connected to its preference for monomeric IL-8, but we have merely noted a correlation. The sentence in question reads:

These functional differences are mirrored by the preference of CXCR1 for the monomeric IL-8.

We have amended this sentence to make clear that any underlying cause, if it exists, remains as yet unknown.

Minor Points;

The reviewer made a number of minor points, including a number of typographical errors, and we are grateful to the reviewer for highlighting these. We have followed the suggestion to use the name CXCL8 instead of IL-8 throughout the paper. Figure 5b has been modified to make the position of the sodium site clear. The names CRS1 and CRS2 have been corrected. With regard to the N-terminus of the CXCL8, we have modified the Methods section to make the sequence absolutely clear. We prefer to keep the brief mention of mass spectrometry in the Discussion to underline the fact that our CXCL8 samples are properly folded and intact.

Reviewers' Comments:

Reviewer #1:

Remarks to the Author:

The authors have addressed all of my comments appropriately.